# Bayesian Network Structure Learning Method Based on Causal Direction Graph for Protein Signaling Networks

**DOI:** 10.3390/e24101351

**Published:** 2022-09-24

**Authors:** Xiaohan Wei, Yulai Zhang, Cheng Wang

**Affiliations:** School of Information and Electronic Engineering, Zhejiang University of Science and Technology, Hangzhou 310023, China

**Keywords:** Bayesian network, structure learning, causal direction, protein signaling network

## Abstract

Constructing the structure of protein signaling networks by Bayesian network technology is a key issue in the field of bioinformatics. The primitive structure learning algorithms of the Bayesian network take no account of the causal relationships between variables, which is unfortunately important in the application of protein signaling networks. In addition, as a combinatorial optimization problem with a large searching space, the computational complexities of the structure learning algorithms are unsurprisingly high. Therefore, in this paper, the causal directions between any two variables are calculated first and stored in a graph matrix as one of the constraints of structure learning. A continuous optimization problem is constructed next by using the fitting losses of the corresponding structure equations as the target, and the directed acyclic prior is used as another constraint at the same time. Finally, a pruning procedure is developed to keep the result of the continuous optimization problem sparse. Experiments show that the proposed method improves the structure of the Bayesian network compared with the existing methods on both the artificial data and the real data, meanwhile, the computational burdens are also reduced significantly.

## 1. Introduction

The protein signaling network describes the interactions between different types of proteins. It is critical for discovering the unknown effects or even unobserved components in molecular biology [1]. However, due to the complexity of the signaling pathways, it is difficult to achieve this goal simply via biological experiments [2]. Fortunately, the network can be obtained in another way by analyzing the variables, such as protein or phospholipid expression levels in the cells [2]. Therefore, the construction of protein signaling networks becomes an important issue in the current research of bioinformatics. Since Bayesian network is an effective tool to construct such multivariate relationships [3], the construction of protein signaling networks is equivalent to the structure learning problem of Bayesian networks.

Most of the existing Bayesian network structure learning methods can be divided into two categories in general, namely constraint-based structure learning methods and score-based structure learning methods [3]. Constraint-based methods focus on the conditional independence among the observation variables [4] and select the graph structures that can best express the conditional independence in data [5]. On the contrary, in score-based approaches the structure learning will be taken as model selection problems [6], and the results will be given by optimization problems with loss functions and constraints.

However, there are two major problems in the practice of building protein signaling networks using these classic Bayesian network methods. First, the network structures representing correct joint probabilities do not guarantee the correct causal directions. Bayesian networks with different causal structures may represent identical joint probabilities and conditional probabilities [7]. Second, the structure learning of Bayesian networks can be taken as a combinatorial optimization problem with high computational complexity. The searching space of the structure graph is $O(2^{d})$, where *d* is the number of observation variables or the number of nodes in the graph, so it is an NP-hard problem [8].

Many studies have been conducted in recent years to address both of the above issues. On the topic of Bayesian network with causality, structure learning methods developed in [9,10] assume that the users can obtain additional data by controlling some of the variables during structure learning, which is unfortunately infeasible when building protein signaling networks; On the other hand, the methods with pure observation data may have extra assumptions on the data distributions. As an example, ref. [11] assumes that the observations are produced by a linear structural equation model with non-Gaussian additive noise. These assumptions are not consistent with that of the protein signal data. Score-based methods with black box neural network models are also developed in this field [12,13]. The results of these neural network-based methods are largely dependent on the initial positions in the searching spaces as well as many other hyper-parameters of the models; moreover, a huge amount of calculations are required. So, on the topic of reducing the complexity of the structure learning, improved methods with greedy search policies [14,15] and heuristic policies [16] can be found in the literature on Bayesian network structure learning to accelerate the searching. A very different trick is used in the work of [17], where the combinatorial optimization problem of the network structure learning is converted to a continuous optimization problem to reduce the computational complexity. This technique will be leveraged in the proposed method in this paper.

Thus, a new Bayesian network structure learning algorithm is proposed in this paper to construct the protein signaling networks. First, a causal direction matrix can be calculated by the algorithm in [18], since the relationships between any two variables in protein signaling networks are very likely to be non-linear. Second, the causal direction matrix is used as one of the constraint functions in a continuous optimization problem which minimizes the fitting loss of a structural equation model [17]. Finally, a pruning method is developed to keep the structure matrix sparse. In the experiment section, we show that the method proposed in this paper outperforms the existing methods in accuracy and computational efficiency.

In the following part of this paper, preliminaries will be given in Section 2. The details of the proposed method will be given in Section 3, and experimental results will be demonstrated in Section 4.

## 2. Preliminaries

### 2.1. Causal Direction Inference

In Bayesian network structure learning, the correct causal directions are crucial, since many graphs with edges in different causal directions probably represent identical joint probabilities. In recent years, progress has been made in inferring the causal directions between two variables [19]. The IGCI method in [18] will be used in this work, since its assumptions on the variables are very similar to that in the protein signaling networks.

The very basic idea of the IGCI algorithm is that a non-linear transfer function will change the distribution of the output variable, meanwhile, that of the input variable will stay unchanged [18]. Mathematically speaking, for a nonlinear function f(·), whose input variable is represented by *x* and the output variable is represented by *y*. The concept of *x* causes *y* can be written as y=f(x). Let the probability density function of *x* be $p_{x}\left(x\right)$, and $g\left(\cdot \right)=f^{-1}\left(\cdot \right)$, then the probability density function of the effect variable *y* can be written as:(1)$$p_{y}\left(y\right)=p_{x}\left(g\left(y\right)\right)\left|g^{\prime }\left(y\right)\right|$$

From (Equation 1) we can tell that the correlation between $p_{y}$ and $|g^{\prime }\left(y\right)|$ should be more significant than that between $p_{x}$ and $|f^{\prime }\left(x\right)|$. This asymmetry property can be used to infer the causal directions between two variables. If the variable *x* causes the variable *y*, we can obtain
(2)$$\int \log_{}\left|f^{\prime }\left(x\right)\right|p_{x}\left(x\right)dx<\int \log\left|g^{\prime }\left(y\right)\right|p_{y}\left(y\right)dy$$

Equation (Equation 2) can be taken as the calculation of comparing the mathematical expectations of log$|f^{\prime }\left(x\right)|$ and log$|g^{\prime }\left(y\right)|$. In a Bayesian network, since we do not know the causal directions, $x_{i}$ and $x_{j}$ will be used instead of *x* and *y*. Take two nodes $x_{i}$ and $x_{j}$ (where 1≤i,j≤d and i≠j). $x_{i}$ and $x_{j}$ are two column vectors in the entire data matrix *X*, and each column vector is composed of *m* observation values. Then $x_{i}^{\left(k\right)}$ denotes the *k*th element in the vector $x_{i}$ after listing all the observations of $x_{i}$ in ascending order, and $x_{j}^{\left(k\right)}$ denotes the *k*th element of the $x_{j}$ after listing all the observation of the variable $x_{j}$ in ascending order (1≤k≤m). The causal direction of these two nodes can be calculated in an equivalent way as follows.
(3)$$C_{x_{i}\to x_{j}}=\frac{1}{m-1}\sum_{k=1}^{m-1}\log\left|\frac{x_{j}^{\left(k+1\right)}-x_{j}^{\left(k\right)}}{x_{i}^{\left(k+1\right)}-x_{i}^{\left(k\right)}}\right|$$
(4)$$C_{x_{j}\to x_{i}}=\frac{1}{m-1}\sum_{k=1}^{m-1}\log\left|\frac{x_{i}^{\left(k+1\right)}-x_{i}^{\left(k\right)}}{x_{j}^{\left(k+1\right)}-x_{j}^{\left(k\right)}}\right|$$

If $C_{x_{i}\to x_{j}}$ is less than $C_{x_{j}\to x_{i}}$, the causal direction is $x_{i}$ causing $x_{j}$, otherwise, the causal direction is $x_{j}$ causing $x_{i}$. Since the denominator in (Equation 3) and (Equation 4) could be zero if there are identical observations, repeated values will be handled in the algorithm described in Section 3.

### 2.2. Acyclic Graph Constraint

A Bayesian network can be described as a directed acyclic graph, so acyclic is another constraint to constructing the structure. Whether there is a cycle in the graph can be told by the adjacency matrix *W*. In addition, since continuous optimization will be discussed in this paper, we assume that the adjacency matrix *W* is weighted, so $W\in R^{d\times d}$. This is also more general than the unweighted cases. Note that the final result of the Bayesian structure learning is $W\in {\left\{0,1\right\}}^{d\times d}$.

For $k\in N^{+}$ and 1≤i,j≤d, the (i,j)-th element of the power matrix $W^{k}$ is the sum of the products of the weights along all *k*-step paths from node *i* to node *j*. The trace function tr(·) is the sum of the diagonal elements of a matrix. So if the graph is acyclic, the trace of all $W^{k}$ for all *k* should be zero, and the acyclic constraint has to ensure that $tr(W^{k})$ is equal to 0 when *k* is equal to any positive integers.

However, it is impossible to testify all cases since the value of *k* can be infinite, so the problem can be solved by a mathematical trick by using the Taylor expansion on the exponential matrix of *W* as Equation (Equation 5) describes.
(5)$$tr\left(e^{W}\right)=tr\left(I\right)+\mathrm{tr}\left(W\right)+\frac{1}{2!}tr\left(W^{2}\right)+\ldots \geq d$$

If the directed graph described by the matrix *W* is acyclic, $tr(e^{W})$ must be equal to *d*, which is the size of the matrix *W* and the number of nodes as well. However, in some cases, the values in the *W* matrix can be negative, so the Hadamard product is used to avoid negative weights. The final acyclic constraint can be written as
(6)$$\mathrm{tr}\left(e^{W\circ W}\right)-d=0$$

As long as the above constraints are satisfied, we consider the causal graph we constructed to be a directed acyclic graph.

### 2.3. Loss Function Based on Structural Equations

Structural equation modeling (SEM) is a statistical method for analyzing the relationships between variables based on their covariance matrices, and it plays an important role in understanding the relationships between observed variables and latent variables [20]. The random vector $X=\left(x_{1},\cdots ,x_{d}\right)$ is the normalized data that we observe. $X\in R^{n\times d}$ represents the observed data matrix *d* observed variables and there are *n* samples for each variable. The coefficient matrix *W* has the same size with the weighted adjacency matrix. The corresponding structure equation can be written as follows.
(7)X=XW+ϵ

$\epsilon =\left(\epsilon_{1},\cdots ,\epsilon_{d}\right)$ is the fitting error matrix. As Equation (Equation 8) shows that, the least squares loss function of all the fitting errors can be used as the loss function of the structure learning.
(8)$$F\left(W\right)=\frac{1}{2n}{∥X-XW∥}_{2}^{2}$$

In (Equation 8), $\left|\right|\cdot {\left|\right|}_{2}$ denotes the matrix Frobenius norm. Since the minimization of the least squares loss function is shown to recover the true directed acyclic graph with a high probability on finite samples [21], if the minimization of the above equation holds, then the graph constructed from the final derived matrix *W* is the causal graph we need.

## 3. Methods

As described in Section 1, a continuous optimization with causal direction matrix constraints is proposed in this work to construct the structure of the Bayesian network. In this section, the computation of the causal direction matrix is described first, the optimization problem, as well as its numerical solving algorithm, is developed next, and the pruning method is shown at the end of this section.

### 3.1. Causal Direction Matrix

The causal direction constraint matrix *G* is calculated in this section. If $x_{i}$ cause $x_{j}$, the (i,j)th element of the matrix is 1, otherwise, it is set to be zero. Note that this causal relationship can be either direct or indirect. This matrix is able to avoid edges with incorrect causal direction if the following equation holds.
(9)$$||W\circ \bar{G}{\left|\right|}_{2}^{2}=0$$

In (Equation 9), $\bar{G}$ is a complementary matrix of *G* where 1 turns into 0 and 0 turns into 1. When (Equation 9) holds, all the causal directions in the matrix *W* are identical to that in the matrix *G*. The elements in the causal direction matrix *G* can be calculated by (Equation 3) and (Equation 4). However since the denominator of (Equation 3) and (Equation 4) may be zero if duplicated elements occur [19], the actual calculation will be performed as in Equations (Equation 10) and (Equation 11).
(10)$$C_{x_{i}\to x_{j}}=\frac{1}{\sum_{k=1}^{{\tilde{m}}_{i}-1}n_{i}^{\left(k\right)}}\sum_{k=1}^{{\tilde{m}}_{i}-1}n_{i}^{\left(k\right)}\log\left|\frac{{\tilde{x}}_{j}^{\left(k+1\right)}-{\tilde{x}}_{j}^{\left(k\right)}}{{\tilde{x}}_{i}^{\left(k+1\right)}-{\tilde{x}}_{i}^{\left(k\right)}}\right|$$
(11)$$C_{x_{j}\to x_{i}}=\frac{1}{\sum_{k=1}^{{\tilde{m}}_{j}-1}n_{j}^{\left(k\right)}}\sum_{k=1}^{{\tilde{m}}_{j}-1}n_{j}^{\left(k\right)}\log\left|\frac{{\tilde{x}}_{i}^{\left(k+1\right)}-{\tilde{x}}_{i}^{\left(k\right)}}{{\tilde{x}}_{j}^{\left(k+1\right)}-{\tilde{x}}_{j}^{\left(k\right)}}\right|$$

Let ${\tilde{x}}_{i}^{\left(k\right)}$ ( $1\leq k\leq {\tilde{m}}_{i}$) be the *k*th largest value after removing all the duplicate elements and ${\tilde{m}}_{i}$ is the number of different values in the vector of variable $x_{i}$, and $n_{i}^{\left(k\right)}$ denotes the number of occurrences of ${\tilde{x}}_{i}^{\left(k\right)}$ in the original dataset. So (Equation 3) can be replaced by (Equation 10), and similarly, (Equation 4) can be replaced by (Equation 11).

The matrix *G* is initialized as a zero matrix of d×d. Once the results of $C_{x_{i}\to x_{j}}$ and $C_{x_{j}\to x_{i}}$ are calculated, the (i,j)th element in the corresponding position of the causal direction matrix *G* can be determined by the Equation (Equation 12).
(12)$$G_{ij}=I\left(C_{x_{i}\to x_{j}}<C_{x_{j}\to x_{i}}\right)$$

I(·) is the indicator function. $G_{ij}=1$ means that the causal direction is from $x_{i}$ to $x_{j}$, otherwise $G_{ij}=0$ means that the causal direction from $x_{j}$ to $x_{i}$. Note again that this causal direction could be either direct or indirect. The pseudo-code of the computation of the causal direction matrix is described in the Algorithm 1.
**Algorithm 1: **Causal Direction Matrix
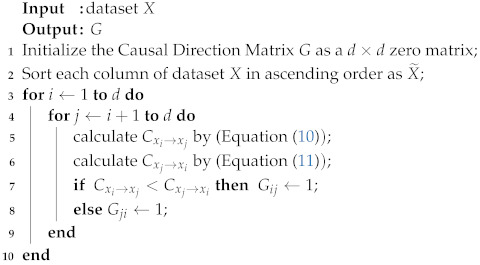


   The matrix is a complementary symmetric boolean matrix with $G={\bar{G}}^{T}$.

### 3.2. Optimization Problem for Structure Learning

A least square fitting loss of an SEM (structural equation modeling) is used in this section to construct an optimization problem for structure learning. At the same time, an $l_{2}$-regularization is added to restrict the absolute values of the weights. So the optimization problem can be written as follows.
(13)$$\begin{array}{c} \begin{array}{c} \min_{W\in R^{d\times d}}S\left(W\right)=\frac{1}{2n}{∥X-XW∥}_{2}^{2}+\lambda {∥W∥}_{2}^{2} \end{array}\\ \begin{array}{cc} s.t.\;\;\;\mathrm{tr}(e^{W\circ W})-d & =0\\ ∥W\circ \bar{G}{∥}_{2}^{2} & =0 \end{array} \end{array}$$

There are two constraints in this optimization problem. The first one is the acyclic constraint, which ensures that the directed graph constructed by the weighted adjacency matrix *W* is acyclic. The details of the acyclic constraint have been given in Section 2.2. The second one is the causal direction matrix constraint, which ensures that the prediction graph has identical causal directions with the causal direction matrix *G*. The computation of *G* has been described in detail in Section 3.1.

To solve the above optimization problem, i.e., to minimize the objective function with a given set of specified equation constraints, the Lagrange multiplier method and gradient decent method are used here to solve this problem.
(14)$$\min_{W\in R^{d\times d}}L\left(W,\alpha ,\beta \right)=S\left(W\right)+\alpha h\left(W\right)+\beta g\left(W\right)$$
where
(15)$$h\left(W\right)=\mathrm{tr}\left(e^{W\circ W}\right)-d$$
(16)$$g\left(W\right)={∥W\circ \bar{G}∥}_{2}^{2}$$

All the items in the above target function are derivable and their gradients can be written as follows.
(17)$$\nabla S\left(W\right)=\frac{1}{n}X^{T}\left(X-XW\right)+2\lambda W$$
(18)$$\nabla h\left(W\right)={\left(e^{W\circ W}\right)}^{T}\circ 2W$$
(19)$$\nabla g\left(W\right)=2W\circ \bar{G}\circ \bar{G}$$

Then the gradient with respect to the above optimization problem L(W,α,β) can be written as follows.
(20)$$\nabla L\left(W,\alpha ,\beta \right)=\frac{1}{n}X^{T}\left(X-XW\right)+2\lambda W+\alpha \left({\left(e^{W\circ W}\right)}^{T}\circ 2W\right)+\beta \left(2W\circ \bar{G}\circ \bar{G}\right)$$

So this optimization problem can be solved by many mathematical tools, an algorithm based on gradient descent is written in Algorithm 2 as follows.
**Algorithm 2:** Gradient Decent Algorithm for Optimization of Problem (Equation 14)
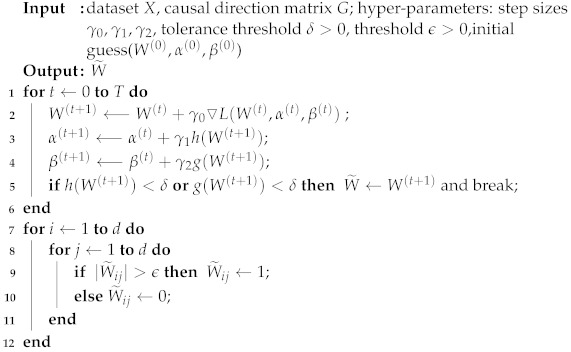


   In the experiment section, the values of the hyper-parameters in Algorithm 2 are set as follows. The step sizes $\gamma_{0},\gamma_{1}$, and $\gamma_{2}$ are all set to be 1, the tolerance threshold δ is set to be $10^{-8}$, and the threshold ϵ is set to be 0.2. $\alpha^{\left(0\right)}$ and $\beta^{\left(0\right)}$, where the initial values of α and β are set to 0, and $W^{\left(0\right)}$ is an all-zero matrix.

### 3.3. Pruning Algorithm

A pruning method is also proposed to make the final result graph more sparse. This method is inspired by the concept of Granger causality [22], which uses the absolute values of the coefficients of a regression model to determine whether one variable is the cause of another variable. A second-order polynomial regression model is constructed for each node with parent nodes. For a parent node in one of these models, if its regression coefficients of the linear term, second order term and cross term are all sufficiently small, this node can be safely removed in this regression model.

The parent node of node $x_{i}$ are denoted by $x_{pa(k,i)}$, where 1≤k≤l. Since the node may have multiple parents, *k* represents the *k*th parent of the node and *l* represents the total number of parents. We perform a second-order polynomial expansion on these parent nodes and fit them into a linear regression model, which leads to the following Equation (Equation 21).
(21)$$x_{i}=\sum_{k=1}^{l}a_{k}x_{pa(k,i)}+\sum_{\begin{array}{c} k=1:l-1\\ h=k+1:l \end{array}}b_{kh}x_{pa(k,i)}x_{pa(h,i)}+\sum_{k=1}^{l}c_{k}x_{pa(k,i)}^{2}$$

In practice, first we expand the data matrix to contain the quadratic terms and cross terms, and then any linear regression algorithms can be applied. The pruning algorithm is written in Algorithm 3.
**Algorithm 3:** Pruning Algorithm
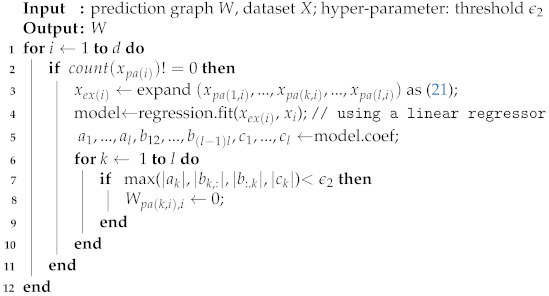


The threshold $\epsilon_{2}$ in the Algorithm 3 is set to be 0.6 in the experiment section, and $W_{pa(k,i),i}$ represents edge from the *k*th parent node of node $x_{i}$ to the node $x_{i}$.

Finally, the Bayesian structure learning procedure with causal direction graph and continuous optimization is summarized in Algorithm 4 and is abbreviated to CO-CDG in the following discussions in this paper. Note that all the hyper-parameters are omitted for clarity and simplicity.
**Algorithm 4:** The CO-CDG Algorithm**Input**:   dataset *X*
**Output**:  *W*
1Causal Direction Matrix *G* = Algorithm1(*X*);2Prediction Graph *W* = Algorithm2(*X*, *G*);3Final Prediction Graph *W* = Algorithm3(*X*,*W*);

## 4. Experiment Results

In the experiment section, we focus on the correctness of the learned structures and the running time of the proposed algorithm. The performances of the proposed method are compared with the FGES (fast greedy equivalence search) algorithm in the reference [15], the neural network-based DAG-GNN algorithm in the reference [12], the NoTears algorithm in the reference [17], the reinforcement learning-based RL-BIC algorithm and RL-BIC2 algorithm in the reference [13].

The link to the code packages of the above comparison algorithms are listed as follows:FGES algorithm: https://github.com/bd2kccd/py-causal accessed on 22 February 2022;NoTears algorithm: https://github.com/xunzheng/notears accessed on 30 July 2021;DAG-GNN algorithm: https://github.com/fishmoon1234/DAG-GNN accessed on 24 February 2022;RL-BIC and RL-BIC2 algorithms: https://github.com/huawei-noah/trustworthyAI/tree/master/research accessed on 11 May 2021.

The structural Hamming distance (SHD) [23] is used to measure the correctness of the structures. SHD can be calculated as
(22)$$\mathrm{SHD}=N_{E}+N_{M}+N_{R}.$$

In (Equation 20), $N_{E}$ represents the number of redundant edges that need to be removed, $N_{M}$ represents the number of missing edges that need to be added, and $N_{R}$ represents the number of edges in the opposite direction that need to be reversed. Thus, the structural Hamming distance (SHD) is the total number of operations that are required to convert a predicted graph to the true graph. The operations include adding, deleting, and direction reversing of the edges. Lower structural Hamming distance represents a better structure when the true structure is available.

In the following Sections, both artificial data and real data are used to demonstrate the performance of the proposed method.

### 4.1. Experiments on Artificial Data

The generation of the artificial data is briefly described here, more details can be found in [13]. Given the number of nodes *d*, a d×d upper triangular matrix is generated randomly as the binary adjacency matrix of the graph. The elements of the upper triangular matrix are sampled from a Bernoulli distribution with p=0.5. If node $x_{i}$ has multiple parent nodes, a second-order polynomial expansion, which is identical to Equation (Equation 21) in Section 3.3 of this paper, is used. The coefficients in (Equation 21) are either set as 0 or randomly sampled from a uniform distribution [−1.5,−0.5]∪[0.5,1.5]. The probability of a coefficient equal to 0 is 50%; thus, if a parent node has no contribution to the child node due to 0 coefficients being generated, the value 1 in the corresponding position in the true binary adjacency matrix should be changed to 0. Next, noise samples are generated from a standard Gaussian distribution. Finally, a graph with *d* nodes is generated and the data set has 5000 samples for each node. In order to avoid the influence of the outliers, the samples are sorted in ascending order by their absolute sums, and then the first 3000 samples are used for the following experiments.

In the following experiments in this section, the SHDs of the proposed method will be evaluated under different data lengths and different number of nodes. In the experiments in Figure 1, artificial data sets are generated in which the number of nodes range from 5 to 14. The results show that with the increase in the number of nodes, the SHDs of the proposed algorithm are significantly smaller than the other algorithms.

In Table 1, the running times of the above algorithms are also evaluated and the experiment settings are the same. The results show that the running time of the proposed algorithm is significantly smaller than the other algorithms as the number of nodes increases. The running time of the FGES algorithm is smaller than that of the proposed algorithm in some cases; however, the corresponding SHD values are much bigger than that of the proposed algorithm.

In the experiments in Figure 2, the number of nodes is set to be 11, which is equal to that in the real data set. 100 data points are used at first and 10 data points are added each time to increase the experiment data length. Thus, there are 291 recorded results in a single curve in the following figure. Note that the order of the data points is random. As can be seen from Figure 2, with the increase in the length of data, the SHDs of the proposed algorithm are significantly smaller than the other algorithms as well.

Next in Table 2, the running times of the methods are listed. The number of nodes is set to be 11, and the data length is set to be 3000; these settings are identical to the last group experiment in Figure 2. Not surprisingly, the proposed method has a high computational speed.

### 4.2. Experiments on Protein Signaling Network Data

Experiments in this section are performed on a real data set of protein signaling network in the work of [2]. There are 11 signaling nodes in this data set, each of which represents a phosphorylated protein molecule in the research of the human primary T cell signaling pathway. A causal graph, which can be taken as the ground truth, has been constructed by classical biochemistry and genetic analysis over the past two decades, just as shown in the Figure 3. There are 20 edges in this graph, where 17 of them are high-confidence causal edges, and the rest are low-confidence causal edges. This study relies solely on the data to obtain the results without reference to the biological knowledge behind the data.

There are 14 sub-datasets with respect to 14 different biochemical experiments. The details of data collection can be found in [2]. Note that the number of data points is different for each sub-data set, ranging from 723 to 917.

#### 4.2.1. Comparison of SHDs of Different Algorithms

The results of the algorithms on all samples of the 14 sub-datasets are listed in Table 3. The proposed method gives the best results in more than half of the sub-datasets. Additionally, the average SHD is also smaller than all of its counterparts.

Note that the results of all the algorithms, including the proposed one, are sensitive to the order of variables in the data set, which is out of our expectation since it cannot be well explained from the perspective of mathematical formulations, at least for the proposed method here. Furthermore, from the results, there are no significant clues to answer the questions, such as in what order will the results be better, and why. Thus, this will be one of our future works.

A group of experiments on different data lengths is also demonstrated in Figure 4. The experiment settings are similar to that in the experiments of Figure 2 in the section of the artificial data. The results of the third sub-dataset are drawn in Figure 4. There are 911 data points in the third sub-dataset. A total of 100 data points are used at first and 10 data points are added each time to enlarge the experiment data set, and there are 82 recorded results in a single curve here. The proposed method outperforms the existing method on most occasions.

In Table 4, the binary values in the elements of adjacency matrices are taken as the results of a binary classification problem. Thus, the metrics such as accuracy, precision, recall and F1 score are evaluated to provide more details. The experiment settings for Table 4 are the same with that of the points in the last column of Figure 4. From Table 4, it is also clear that the algorithm proposed in this paper is capable of achieving better results under the metrics of binary classification.

#### 4.2.2. Comparison of Running Time

Similarly, the average running times of the algorithms on 14 sub-datasets are listed in the Table 5. The average running time of the proposed method is significantly shorter than the existing methods.

#### 4.2.3. Analysis of the Result Structures

The proposed algorithm gives satisfactory results in a large part of the graph. Just as shown in Figure 5, there are seven nodes and nine edges in the subgraph of ground truth (left subfigure). The result graph (right subfigure) gives seven correct edges. Only one edge is missing and one edge has an incorrect causal direction.

On the other hand, in the example in Figure 6, two out of three edges given by the proposed method have incorrect causal directions. This result indicates that the SHDs can be further improved if the Algorithm 1 in this paper gives better guesses of causal directions.

Note that the sub graph in Figure 5 is in the right box of Figure 3 and the sub graph in Figure 6 is in the left box of Figure 3.

## 5. Conclusions

In this paper, a continuous optimization algorithm with causal direction matrix constraints is proposed to learn the structure of the Bayesian network for protein signaling variables. The results of the proposed algorithm outperform the existing methods. The research in protein signaling network and related fields can be effectively accelerated with the help of the structures given automatically by the proposed method. From the perspective of a pure biologist, admittedly, nothing new has been discovered in this paper. However, it is proved that the proposed method is capable of discovering something that we already knew to be true, which indicates that it can be applied to more applications with similar data distributions.

There are two possible future works under the current framework of this paper, which lie in the two constraints of the proposed optimization problem, respectively. First, the accuracy of the causal direction matrix can be further improved, the assumption of the causal direction inference method used in this paper may be not consistent with some of the edges. Second, the acyclic assumption of the Bayesian networks may be also not correct in some cases. Topology with circles cannot be excluded in the field of protein signaling research, so a dynamic Bayesian network should be a more general choice. In addition, from the perspective of data, multi-modal data are also a potential challenge in the related field [24].

## Figures and Tables

**Figure 1 entropy-24-01351-f001:**
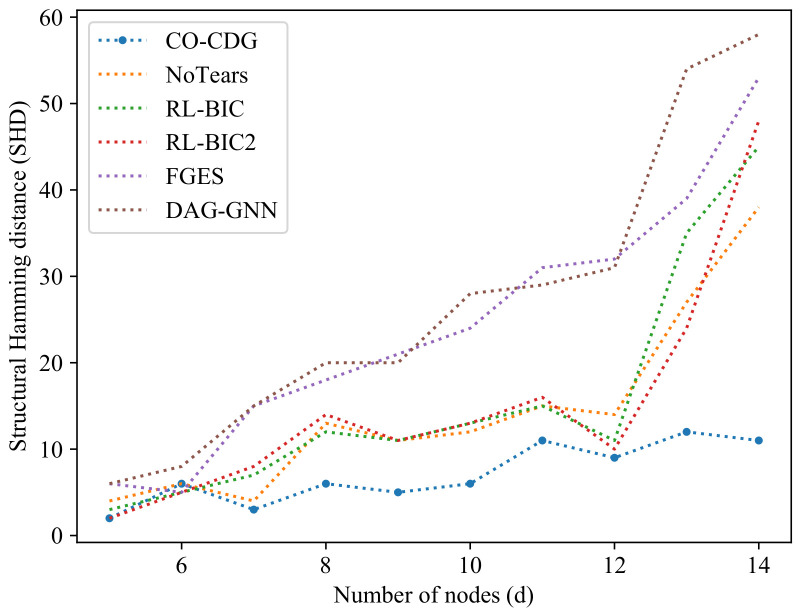
Comparison of the structural Hamming distance of the results on artificial datasets with different numbers of nodes.

**Figure 2 entropy-24-01351-f002:**
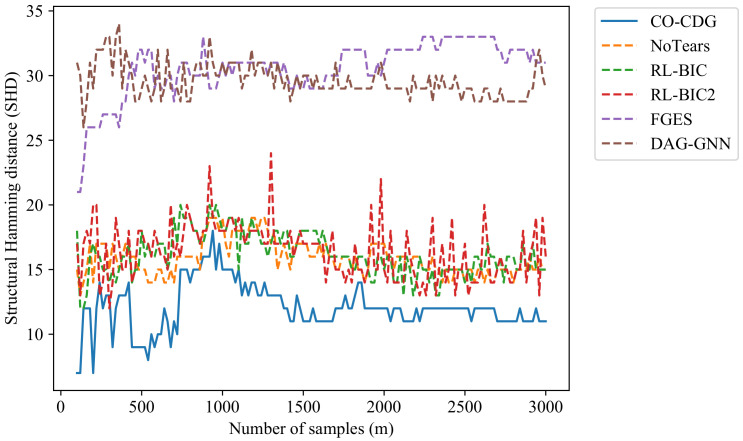
Comparison of the structural Hamming distance of the results on artificial datasets with different data lengths.

**Figure 3 entropy-24-01351-f003:**
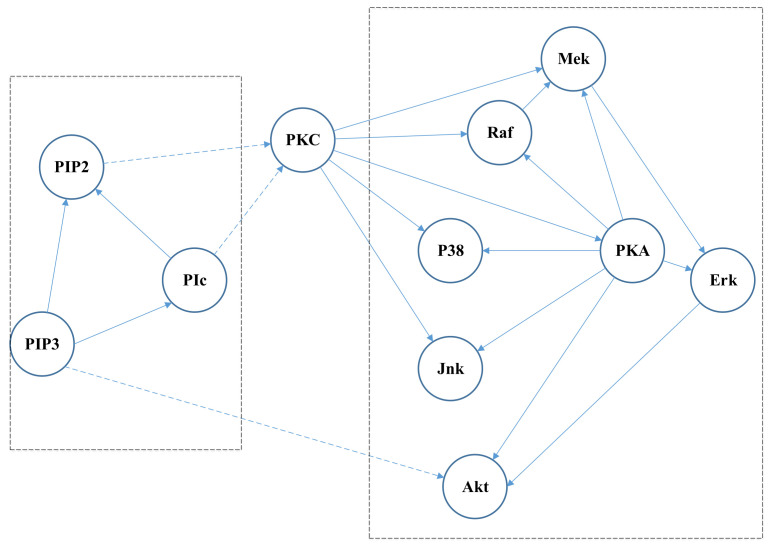
Protein signaling network in the research of human primary T cell signaling pathway.

**Figure 4 entropy-24-01351-f004:**
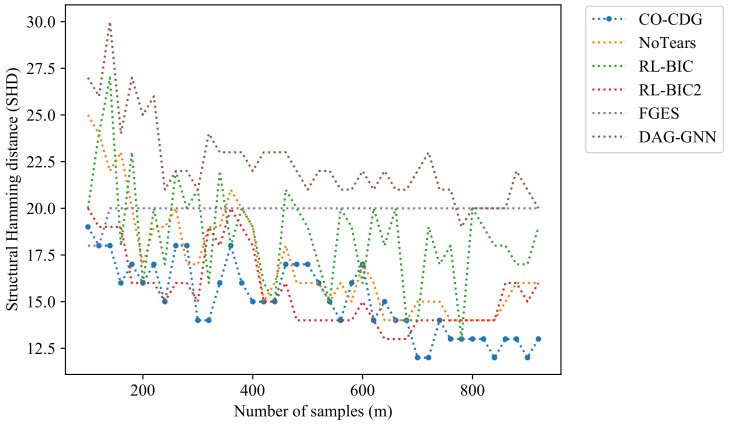
Comparison of SHDs of different data lengths on the real data set.

**Figure 5 entropy-24-01351-f005:**
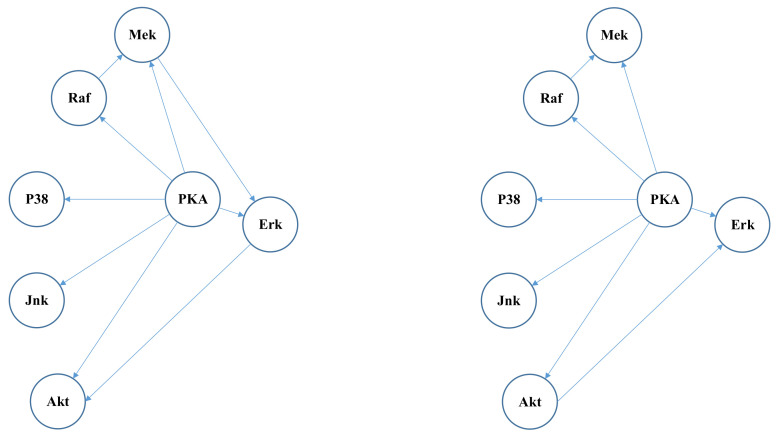
A satisfactory example: ground truth graph (**left**) and result graph given by the proposed algorithm (**right**).

**Figure 6 entropy-24-01351-f006:**
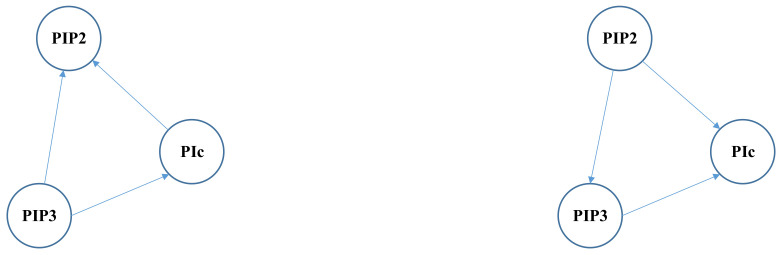
A bad example: ground truth graph (**left**) and result graph given by the proposed algorithm (**right**).

**Table 1 entropy-24-01351-t001:** Comparison of running time (s) on artificial datasets with different number of nodes.

Number of Nodes	CO-CDG	NoTears	FGES	DAG-GNN	RL-BIC2	RL-BIC
5	0.87	0.69	1.21	45.87	52.47	54.88
6	0.83	0.60	1.18	43.85	62.44	64.03
7	1.58	2.36	2.27	46.20	72.54	78.10
8	6.80	8.77	2.56	47.01	77.15	77.49
9	2.34	2.74	3.41	48.63	98.42	99.74
10	2.84	4.34	3.40	50.62	112.84	114.95
11	3.41	5.45	4.38	56.64	134.44	134.62
12	4.00	3.23	4.46	52.58	153.30	157.65
13	10.01	22.1	4.49	55.67	209.09	205.14
14	12.98	28.73	4.62	59.78	268.55	285.14

**Table 2 entropy-24-01351-t002:** Comparison of running times on artificial datasets

Algorithm	CO-CDG	FGES	NoTears	DAG-GNN	RL-BIC2	RL-BIC
Time (s)	2.65	4.35	5.66	55.76	252.85	325.03

**Table 3 entropy-24-01351-t003:** Structural Hamming distance results for multiple algorithms on 14 subsets of the real data.

DataSet	CO-CDG	NoTears	RL-BIC	RL-BIC2	FGES	DAG-GNN
1	15	14	19	16	20	19
2	16	13	19	16	21	22
3	13	16	19	16	20	20
4	22	24	21	20	20	25
5	17	17	18	20	21	21
6	18	21	20	19	21	24
7	17	17	20	19	21	23
8	17	15	18	17	21	20
9	19	24	19	24	21	27
10	17	17	18	17	21	19
11	16	19	18	15	20	23
12	21	23	20	23	23	20
13	17	17	16	20	22	22
14	15	18	20	20	22	21
Mean	17.14	18.21	18.93	18.93	21.0	21.86

**Table 4 entropy-24-01351-t004:** Evaluating the adjacency matrix by the metrics of binary classification.

	CO-CDG	NoTears	FGES	DAG-GNN	RL-BIC2	RL-BIC
Accuracy	0.85	0.81	0.83	0.80	0.78	0.82
Precision	0.56	0.44	0.50	0.35	0.29	0.40
Recall	0.45	0.40	0.35	0.25	0.20	0.10
F1 scoce	0.50	0.42	0.41	0.29	0.24	0.16

**Table 5 entropy-24-01351-t005:** Comparison of average running times of the algorithms on 14 subsets of the real data.

Algorithm	CO-CDG	NoTears	FGES	DAG-GNN	RL-BIC2	RL-BIC
Time	3.78	5.52	7.14	15.76	126.31	137.93

## Data Availability

Not applicable.

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
