# Peer review of "Bayesian Network Structure Learning Method Based on Causal Direction Graph for Protein Signaling Networks"

_entropy, 2022, doi:10.3390/e24101351_

Round 1

Reviewer 1 Report

This paper proposes a new continuous optimisation BN structure learning algorithm intended for causal relationship learning between protein signalling variables.

The authors have not clarified whether the pruning technique is sound. Based on their description, it looks like it is not sound. However, it is unclear under what circumstances a true edge will be pruned off when using this technique. The authors should decide how to best provide this information.

After going through the algorithm, I`m having difficulty understanding which components of the algorithm are novel and which components come from other algorithms. For example, the authors have already clarified that the architecture of their algorithm is based on NOTEARS, but it is unclear to what degree the different components are the same. Please clarify, for each component, whether it comes from some other algorithm, whether it is based on some other algorithm, or whether it is completely novel.

The information about which software package was used to test each of the competing algorithms is missing. This is important because the same algorithms are available across different open-source packages, and their running time and accuracy differs depending on the selected library. Please provide this information.

Many algorithms are sensitive to the order in which the variables are present in the data. The authors should test whether this is the case for their algorithm, and clarify in the paper.

The results look good. However, the evaluation is solely based on the SHD score, which is known to be biased in favour of sparse structures. In other words, algorithms that produce very few edges (or even empty graphs) are often incorrectly classified as superior to other algorithms on the basis of the SHD score.  The authors should include the F1 score, which composes of precision and recall, to validate the results. They should also present the average number of edges produced by each algorithm.

The results are said to be based both on synthetic and real datasets. However, it is not clear what process the authors followed to evaluate the real dataset. The authors have used the SHD score for the real case study, but it is not clear how the SHD scores were retrieved since they require access to the ground truth graph. Did the authors used their own hypothetical true structure? Clarification and justification is required.

I found some of the text to be either clumsy or incorrect. For example, the authors state in the abstract that “classic structure learning algorithms of the Bayesian network take no account of the causal relationships between variables”. This is not true since the constraint-based algorithms are already known to operate on causal classes in determining edges and orientation of some of those edges. The authors will have to explain what other non-classic algorithms they refer to that operate on causal relationships, and how they relate to the causal classes of constraint-based learning.

The manuscript is ‘OK’ written. I recommend proof-reading the article and performing minor-to-moderate editing of English language to improve writing where possible.

Reviewer 2 Report

In this paper, the authors describe the computation of causal directions in protein signal networks between any two variables and store them in a graph matrix as one of the constraints of structure learning. A continuous optimization problem is then solved using the fitting losses of the corresponding structural equations as the objective. The directed acyclic prior thus obtained is simultaneously used as another constraint. Moreover, the authors present a pruning method to keep the result of the continuous optimization problem sparse. The experiments conducted and documented show that the proposed method improves the structure of the Bayesian network compared to the existing methods for both artificial data and real data, and significantly reduces the computational cost.  

In general, the construction of protein signal network structure using Bayesian network technology is an important topic for bioinformatics and thus for many applications in medicine, biology, etc. On the positive side, this work is very well written, interesting and relevant, and represents a good contribution to the international research community. Negative and in need of improvement is the related work: the references cited are all excellent and fit well, however do not reflect the state of the art in this fast growing and important field - the only novel related work is from 2020, this needs to be improved, furthermore the graphics need improvement. Due to the importance of this topic, this reviewer is positive, recommends acceptance of this article, and provides a few suggestions below to make the work more interesting for the reader, especially with regard to recent related work:

Learning causal relations is of high interest, especially
within the biomedical domain. The authors demonstrate the applicability of their method by reconstructing a Protein Signaling Network from unstructured data (Section 4.2). However, the data set is rather small. Protein Networks usually consist of  thousands of proteins. Thus, in addition to Table 1, the authors should analyze the  computational demand while increasing the dimension of the data set. Note, the proposed method may be of high interest for the bioinformatics community. The authors could elaborate possibilities to extend their method for the multi-modal case, a good reference for this is: Towards Multi-Modal Causability with Graph Neural Networks enabling Information Fusion for explainable AI. Information Fusion, 71, (7), 28-37, doi:10.1016/j.inffus.2021.01.008.

Reviewer 3 Report

The paper entitled : Bayesian Network Structure Learning Method Based on Causal Direction Graph for Protein Signaling Networks discusses the problem of Bayesan network for signaling on protein relations.

The paper applies the Bayesan theory to the signaling system limiting to only one example which is human primary T cell signaling pathway.

I expect the Authors to define : What is the discovery presented in this paper ?

The paper is of highly general character. The application is limited to one example.

However the general question is :

What is purpose and the goal for this signaling.

Causality is taken into consideration. However the goal is not defined.

The expression saying : „optimization with causal di- 137 rection matrix constraints is proposed in this work to construct the structure of Bayesian 138 network” does not define the aim of optimisation. If it is limited to mathematical matrix optimisation the question is how to interpret this operation on protein networking.

The paper is interesying as mathematical discussion however I do not see any relations to the living organism strategy.

The defintion of the purpose of the protein network action is missing.

I may suggest the search for the answer to the question – What is the main goal of the living organism from the point of view of protein signalling ?

Round 2

Reviewer 1 Report

The authors have addressed my comments satisfactorily.

Author Response

Thank you for your comments.

Reviewer 3 Report

I do not understand the sentence below

In addition, from the perspective 331 of data, multi-modal data is also a potential challenge in the related field. [? ] X.W

I still do not see the question (what protein system ?) the Authors try to answer
